# Cross-Talk between N6-Methyladenosine and Their Related RNAs Defined a Signature and Confirmed m6A Regulators for Diagnosis of Endometriosis

**DOI:** 10.3390/ijms24021665

**Published:** 2023-01-14

**Authors:** Xiaotong Wang, Xibo Zhao, Jing Wang, Han Wu, Yan Cheng, Qiuyan Guo, Tian Liang, Guangmei Zhang

**Affiliations:** Department of Gynaecology, The First Affiliated Hospital of Harbin Medical University, Harbin 150001, China

**Keywords:** endometriosis, m6A regulators, network, diagnosis, immune microenvironment

## Abstract

An RNA modification known as N6-methyladenosine (m6A) interacts with a range of coding and non-coding RNAs. The majority of the research has focused on identifying m6A regulators that are differentially expressed in endometriosis, but it has ignored their mechanisms that are derived from the alterations of modifications among RNAs, affecting the disease progression primarily. Here, we aimed to investigate the potential roles of m6A regulators in the diagnostic potency, immune microenvironment, and clinicopathological features of endometriosis through interacting genes. A GEO cohort was incorporated into this study. Variance expression profiling was executed via the “limma” R package. Pearson analysis was performed to investigate the correlations among 767 interacting lncRNAs, 374 interacting mRNAs, and 23 m6A regulators. K-means clustering analysis, based on patterns of mRNA modifications, was applied to perform clinical feature analysis. Infiltrating immune cells and stromal cells were calculated using the Cibersort method. An m6A-related risk model was created and supported by an independent risk assay. LASSO regression analysis and Cox analyses were implemented to determine the diagnostic genes. The diagnostic targets of endometriosis were verified using PCR and the WB method. Results: A thorough investigation of the m6A modification patterns in the GEO database was carried out, based on mRNAs and lncRNAs related to these m6A regulators. Two molecular subtypes were identified using unsupervised clustering analysis, resulting in further complex infiltration levels of immune microenvironment cells in diversified endometriosis pathology types. We identified two m6A regulators, namely METTL3 and YTHDF2, as diagnostic targets of endometriosis following the usage of overlapping genes to construct a diagnostic m6A signature of endometriosis through multivariate logistic regression, and we validated it using independent GSE86534 and GSE105764 cohorts. Finally, we found that m6A alterations might be one of the important reasons for the progression of endometriosis, especially with significant downregulation of the expressions of METTL3 and YTHDF2. Finally, m6A modification patterns have significant effects on the diversity and complexity of the progression and immune microenvironment, and might be key diagnostic markers for endometriosis.

## 1. Introductions

Endometriosis (EM) is a common condition from which women suffer [1,2]. Typically, it is described as the presence of functional endometrial tissues implanted in areas other than the uterine body, such as the ovaries, peritonea, and deep infiltrations. Although categorized as benign, EM exhibits numerous biological behaviors similar to those of malignancies, including the invasion of adjacent tissues and the induction of tissue remodeling [3,4]. The concept of “EM-associated infertility” was established years back [5], emphasizing that the occurrence of infertility among EM patients was significantly higher than that of the non-disease population; meanwhile, some connections between EM and infertility have been proven, with EM possibly causing infertility or spontaneous abortion by interfering with several pregnancy-related processes, and vice versa [6]. Since the early signs of EM are always non-typical, research based on the etiology and early diagnostic indicators of EM has generated hot spots in recent years.

Epigenetics often refers to DNA methylations, histone modifications, and non-coding RNA-mediated regulations of widespread regulatory mechanisms that modify the biological phenotype without affecting DNA sequences [7,8]. The links of histone alterations and DNA methylations to both the pathophysiology and progression of EM have been proven recently [9]. Post-transcriptional RNA modifications include N6-methyladenosine (m6A), cytosine hydroxylation (hm5C), and N1-methyladenosine (m1A) [10], where m6A is known as the methylation of adenosine (A) at the sixth N position. The m6A methylation process could be catalyzed and stimulated by “writers”, such as METTL3, METTL14, and WTAP, and ceased by “erasers”, such as ALKBH5 and FTO, resulting in a dynamic and reversible modification, while “readers”, such as the YTHDF family and YTHDC family, further recognize and bind the m6A modification sites in RNAs. Numerous studies have revealed that m6A alterations have different regulatory functions, to a great extent, in various types of malignancies and autoimmune or infectious diseases through their involvement in cell proliferation [11], resistance to chemotherapy and radiotherapy [12], and the immune response [13]. However, less research has been focused on the function of RNA methylation in EM, yet it is known as a chronic inflammatory disease.

To completely elucidate the regulatory network of m6A regulators impacted in EM, it is urgently required to understand the cross-talk changes between m6A regulators and these interacting genes. Modifications of m6As by lncRNAs and mRNAs could contribute to the formation of crucial and intricate networks of cellular modulations. Information on these networks might provide essential insights into prospective mechanisms of EM development and present novel therapeutic options for EM. In the present study, genomic alterations were explored in healthy and EM samples from the Gene Expression Omnibus (GEO) dataset for a comprehensive assessment of m6A-associated RNAs. Two distinct molecular isoforms that could be used to predict clinicopathological characteristics and the activities of the immune microenvironment were identified. A diagnostic risk model was further developed for EM patients by integrating the m6A regulators associated with lncRNAs and mRNAs. The results suggest that m6A regulators might be important diagnostic markers and provide new insights into potential mechanisms during the development of EM.

## 2. Results

### 2.1. Summaries of Data In Silico

The entire workflow is displayed in Figure 1. After removing the non-compliant samples, the data, which contained 43 normal endometrial samples, 104 eutopic endometrial samples, and 198 EM samples termed ectopic samples, including peritoneal endometriosis lesions, deep infiltrating endometriosis lesions, sacrouterine ligament lesions, rectovaginal lesions, and ovarian endometrioma, were reserved. A total of 345 samples with a detailed distribution, such as the menstrual cycle phase, are shown in Table 1.

### 2.2. The Landscape of Expression and Diversity of m6A Regulators among Healthy and Disease Samples

To determine whether m6A alterations are relevant to endometriosis, we conducted differential expression analysis of gene expression in all samples. A total of 11 m6A regulators with altered patterns were identified from the landscape of expression among the different sample groups (Figure 2A,B), including six readers, four writers, and one eraser. YTHDF2 and HNRNPA2B1 stood out among the readers as possessing the most significant alteration, and both had considerably reduced expressions in the ectopic lesions. When compared to the other two writers, METTL3 and METTL16 presented a more notable significance than METTL14 and ZC3H13. Among the erasers, FTO showed a significantly increased expression trend between Normal, Eutopic and Ectopic groups with a notable difference among the three groups, whereas ALKBH5 did not. We also discovered that in the ectopic group, erasers had a higher expression, while writers had a significantly downregulated expression compared to the normal group, indicating that the downregulation of m6As may be a key factor in the development of EM. The regulatory interactions of these m6A regulators could manifest as an intricate PPI network (Figure 2C), suggesting multiple cross-links in EM. Additionally, the close transcriptome correlations among writers, readers, and erasers were investigated in the EC, EU, and NM groups, showing that multiple effects were altered in EM (Figure 2D).

### 2.3. The m6As and Their Relative mRNAs Recognized Patterns of Co-Alteration in EMs

After discovering the one-step neighbors in five public databases, we obtained 374 mRNAs interacting with 19 m6A regulators, which are included here to analyze their expression alterations (Figure 3A). Three gene groups of DEGs were analyzed, and we obtained 31 mRNAs in the eutopic–normal group (Figure 3B), 239 mRNAs in the ectopic–normal group (Figure 3C), and 255 mRNAs in the ectopic–eutopic group (Figure 3D), each with FDR < 0.05. SCG2, CTSG, and GPC3 possessed the highest modification extent in all three compared groups. The intersection of these three groups was 25 mRNAs (Figure 3E), meaning a co-alteration of development in EM. Based on the expression levels of all the m6A regulators and the mRNAs (Figure 3F), Pearson correlation coefficients for each mRNA–m6A pair were calculated to construct the m6A co-expression network (MACN) (Figure 3G); a co-expression network was constructed between m6A regulators and mRNAs in the NM, EU, and EC groups. Finally, 15 m6A regulators showed co-alteration in the three groups, with 20 mRNA interactions in the MACN.

### 2.4. Consensus Clustering Analysis for m6A-Related mRNAs Unveiled the Heterogeneity in EMs

To characterize the influence of m6A-related mRNAs in the MACN on the development of EM, we performed unsupervised k-means clustering analysis and calculated the Euclidean distances based on the expression levels of m6A-related mRNAs. According to the delta area, we get a maximum delta area on k = 2, which showed no crossover in the horizontal contrast of Heatmap. So, the value of k = 2 was assessed as the most appropriate number of clusters for further analysis according to the delta area plot and matrix heatmap; thus, cluster1 and cluster2 were identified (Figure 4A,B, Appendix A). Further investigations of these two subtypes revealed different clinicopathological characteristics and expression patterns among patients with EM. As shown in Figure 4C, cluster2 corresponded to a younger age (*p*-value < 0.05) and a more diverse endothelial ectopic pathology type (*p*-value < 0.05) with a higher stage (*p*-value < 0.05) compared to cluster1. However, no significance was shown in the cycle phase, which suggests that hormone factors might not be dominant over others such as genetic alterations. In conclusion, the clustered subtypes could provide a more comprehensive opinion that unveils a significant association with the heterogeneity of EM.

### 2.5. Characteristics of the Immune Microenvironment in EMs Subtypes

As there is emerging immunological evidence relating to EMs, we deconvoluted the mRNA profiles, and investigated the roles of the immune microenvironment, typically the relationship between these two EM subtypes and infiltrating immune cell subpopulations. The results showed different categories of infiltrating immune cells notably between the two subtypes, where total lymphocytes and total dendritic cells were proportionally upregulated in cluster1 (Appendix A), leaving opposite trends for macrophages and mast cells. Amongst them, the proportions of memory B cells, plasma cells, resting memory CD4 T cells, activated memory CD4 T cells, gamma delta T cells, T regulatory cells, resting NK cells, activated NK cells, monocytes, M1 macrophages, M2 macrophages, and resting mast cells were significantly higher in cluster2 than in cluster1 (Figure 4D), underlying the intricate mechanisms forming EM lesions that could be considered for more intensive investigations. In addition, three scores (stromal score, immune score, and estimate score) for both subtypes were evaluated, all indicating higher scores in cluster2 (Figure 4E), representing a higher relative content of stromal cells or immune cells in the immune microenvironment, which might promote heterogeneity and accelerate the progression of EM by stimulating an immunologic reaction and restoring the anatomical relations.

### 2.6. The m6As and Their Relative lncRNAs Recognized Patterns of Co-Alteration in EMs

Next, 767 lncRNAs were screened from the reference genome for differential expression analysis, and 33 lncRNAs with FDR < 0.05 were obtained in the eutopic–normal group (Figure 5A), along with 170 lncRNAs in the ectopic–normal group (Figure 5B), and 186 lncRNAs in the ectopic–eutopic group (Figure 5C). Linc02381 possessed the largest fold change in both the ectopic–normal and ectopic–eutopic groups, while linc00578 was downregulated significantly in the eutopic group. A total of 28 overlapped lncRNAs were selected and differentially expressed in these three groups (Figure 5D,E), and Pearson correlation analysis was performed based on the expression levels of lncRNA–m6A pairs, forming the lncRNA and m6A co-expression network (LACN) that satisfied the threshold, shown as described before (Figure 5F). Finally, 19 m6A regulators showed co-alteration in the three groups, with 20 lncRNAs interactions in the LACN. Furthermore, the random walk algorithm was applied to determine the key m6As, and the mean scores of all nodes were 0.00877193, 0.005347594, and 0.007042254 for the NM, EU, and EC groups, respectively (Figure 5G). The different nodes found in each group indicate dynamic changes among the m6A regulators in EM. Additionally, METTL3, as a vital regulator, scored more remarkably in all three groups and thus might be a potential indicator in EM.

### 2.7. Construction and Validation of an m6A-Related Diagnostic Signature

To explore the diagnostic efficacy of m6A regulators in EM, key m6A regulators from the MACN and key m6A regulators from the LACN were obtained, and an intersection was extracted through differential expression analysis, leaving 14 key m6A regulators (Figure 6A, Appendix A). These m6As were integrated based on LASSO binomial analysis with 10-fold cross-validation (Figure 6B). A total of 11 m6A regulators with non-zero coefficients were screened, namely HNRNPA2B1, METTL3, ZC3H13, RBM15, ELAVL1, LRPPRC, YTHDC2, YTHDF2, FTO, YTHDC1, and YTHDF1, with the minimum lambda value being 0.007609331 (Figure 6C, Appendix A). Subsequently, multivariate logistic regression was applied, including all genes from the LASSO results in the model. The ROC value showed a precise performance in predicting the efficacy of EM (Figure 6D), indicating the vital roles of these m6A regulators in the progression of EM. Aiming for a more concise model, the stepwise regression method was further chosen for screening variables, leaving METTL3, ELAVL1, LRPPRC, YTHDC2, YTHDF2, YTHDC1, and FTO, followed by the confirmation of each value of multicollinearity less than 5 (Figure 6E, Appendix A). The AUC value remained at 0.900. The model was further validated using two independent cohorts, and the AUC remained at 0.875 and 0.984, representing an excellent generalization (Figure 6F). Then, the predicted accuracies of seven m6A regulators were evaluated separately, and the results suggested that METTL3 had the highest AUC value among all writers, with YTHDF2 having the highest AUC value among all readers (Figure 6G, Appendix A). Further, METTL3 was highly correlated with many other regulators in the EC group, with METTL3 and YTHDC2 being the most relevant regulators (Figure 6H, Appendix A). Finally, a nomogram was constructed for risk assessment (Figure 6I). The results showed that YTHDF2 possessed the highest risk weight, followed by METTL3, suggesting that the METTL3–m6A–mRNA/lncRNA–YTHDF2 axis might play a vital role in the progression of EM.

### 2.8. Enrichment Analysis of METTL3-Related Modification Patterns

To investigate the biological responses in the METTL3–m6A modification pattern, the GO and KEGG pathways were explored (Figure 7A). Moreover, using the hallmarks in MSigDB as the gene background, the top three significantly positive and negative hallmarks of biological pathways were assessed through GSEA (Figure 7B, Appendix A). The results showed that several classical immune pathways, such as the TNF-α signaling via NF-kB, inflammatory response, and IL6-JAK-STAT3 signaling-mediated passages, were significantly enriched, suggesting that METTL3-related modification could facilitate immune regulation. The results on the graph show that genes co-expressed with METTL3 are significantly enriched in the signature E2F target, the marker G2M checkpoint, and the MYC targets; therefore, it could be speculated that METTL3 might play a role in regulating the cell cycle, which significantly affects the proliferation of EM cells.

### 2.9. The Experimental Validation of m6A Modification in EMs

After confirming the key m6As, we first employed an m6A quantitative assay to detect the m6A alteration experimentally in EM. We observed that the quantity of m6A modification was increased significantly in normal endometrium species, while it increased the least in the ectopic group (Figure 8A). The RT-qPCR experiment revealed the significant difference in METTL3 and YTHDF2. Although FTO also showed a notably altered expression, there was no significant difference between the ectopic and eutopic groups (Figure 8B). Additionally, the WB experiments revealed similar results, as the difference in METTL3 expression was the most remarkable among them (Figure 8C,D). These findings indicate that METTL3 and YTHDF2 are possibly crucial factors for the formation of EM.

## 3. Discussion

EM is a benign condition with a high incidence in women of reproductive age [14]. Current approaches have limitations in clinical feature-based diagnosis due to the insidious phenotype of EM. Based on extensive research on post-transcriptional modifications, researchers are gradually recognizing the benefits of constructing epigenetic diagnostic models of disease and have confirmed the potential impact of m6A methylation, which is emerging as the most common epigenetic modification, being related to the occurrence and progression of the majority of cancers and other diseases [15]. The majority of m6A regulators were found to have an altered expression in the current study, with “writers” being decreased and “erasers” being upregulated, indicating that the loss of m6A modification might be a potential contributor in EM. Previous research has demonstrated the reverse ability between m6A “writers” and “erasers” on modifications in lncRNAs/mRNAs. m6A “readers” further recognize and bind to methylated lncRNAs/mRNAs and perform diverse functions. As Liu et al. revealed [16], the YTHDF1 complex with YTHDF2 can specifically recognize m6A modifications and thus regulate the stability of lncRNA THOR, impacting the proliferation, migration, and invasion of cancer cells. Moreover, the METTL3–Snail–YTHDF1 axis can promote metastasis in malignant tumors with the modification of EMT-related mRNAs mediated by m6As, which leads to progression [17]. Despite this convincing epigenetic research, little is understood about the function of m6A methylation in EM.

m6A regulators seldom function independently; instead, they control diseases by interacting with genes. Hence, it is crucial to investigate the potential alterations of m6A-interacting genes. Through Pearson analysis, we discovered 25 m6A-regulated mRNAs. Recent studies have demonstrated the impact of SCG2 on the clinical stage, and the influence of macrophage polarization on immunotherapy in colorectal cancer [18,19]. GPC3 [20] has also been proven to be essential in the immune response, yet no reference has been reported in EM. The alterations of m6A-related lncRNAs have been described in depth recently, with abnormal expression typically discovered in EM. Huang et al. observed that the postoperative level of lncRNA-UCA1 was reduced, suggesting that it may function as a diagnostic and prognostic biomarker for EM [21]. ALKBH5 acts as a modification switch of lncRNA SOX2OT and participates in the lncRNA-mediated competitive endogenous RNA model to enhance the molecular stability and exploit the function of SOX2OT, thus affecting progression and drug resistance in glioma [22]. Here, considering the modification of non-coding RNAs by m6As, we analyzed lncRNAs differentially expressed among the EC–EU, EU–NM, and EC–NM groups, in which LINC02381 and LINC00578 showed the largest log2FC values. Concurring with our study, an aberrant expression of LINC02381 was previously confirmed in EM [23], which identified a ceRNA network and verified 28 differentially expressed lncRNAs through analysis of RNA-seq data of EM, followed by RT-qPCR results, confirming that LINC02381 was significantly overexpressed in EM tissues.

The immune microenvironment plays an important role in EM [24], especially in immune cell infiltration and immune dysfunction involved in the progression of EM, as depicted by Wang et al. [25]. Significant diversities were investigated between the two subtypes here at the ratio of 22 immune cells and immune or stromal characteristics. It has been elucidated that T lymphocyte, B lymphocyte, and NK cell infiltration is reduced in EM lesions [26,27,28]. Moreover, previous studies suggested that ectopic endometrial tissue may perform an immunological surveillance function, leading to the formation of chronic inflammation, while a higher stromal score or immune score represents a more stromal or immune component relative to the immune milieu, facilitating inflammation as well as pelvic adhesions, and the estimated score indicates aggregation of the stromal score or immune score in the immune microenvironment, all of which are consistent with our findings.

To determine the critical m6A regulators, LASSO regression was used, followed by further filtration through stepwise regression to screen the variables for a diagnostic model. Finally, seven key m6A regulators were obtained, namely METTL3, ELAVL1, LRPPRC, YTHDC2, YTHDF2, YTHDC1, and FTO. Moreover, independent datasets were used for external validation, making the diagnostic model more robust. A cross-directional analysis revealed that the AUC was the highest for METTL3 in the “writers”, YTHDF2 in the “readers”, and FTO in the “erasers”, indicating remarkable diagnostic efficiency. Consistent with our findings, previous research indicates that METTL3 promotes pre-miR126 maturation via m6A alteration [29], promoting the migration and invasion of endometrial stromal cells in EM [30]. However, YTHDF2 has not been examined in EM. Finally, we experimentally verified the aberrant down-expression of METTL3 and YTHDF2 according to gene and protein levels and found that both m6A regulators posed the highest risk of disease in the ectopic group. The possible enriched downstream targets of METTL3, obtained through enrichment analysis, were E2F targets, the G2M checkpoint, and MYC targets. It has been found that METTL3 activates the G2M checkpoint of the cell cycle through CDC25B mediated by m6A modification, leading to malignant progression in neck squamous cell carcinoma [31]. Additionally, METTL3 could enhance the stability of c-MYC through YTHDF1-mediated m6A modification and promote tumorigenesis in oral squamous cell carcinoma [32]. The expression of the oxidative-phosphorylation-related gene program and the reduction in immune-dependent cell cycle progression are indirectly regulated by METTL3 and YTHDF2, respectively [33], enhancing the high accuracy of the current findings.

## 4. Materials and Methods

### 4.1. Data Pre-Processing

The data used in this study were obtained from the GEO database under the series ID GSE141549, followed by GSE86534 and GSE105764, for subsequent validation. The quantile method was used to normalize the EM-related data of GSE141549, in which samples were grouped into the endometrium, EM, and peritoneum lesions, followed by annotating gene symbols with gene types obtained from GENCODE (version 38). The expression values with duplicate gene symbols were calculated as arithmetic means. Analysis of similarities was applied for three extracted groups, based on a permutation test and rank sum test to determine whether the differences among groups are greater than those within groups, thus verifying notable groupings; *p* < 0.05 was considered significant for the sampling units.

### 4.2. Landscape of Alteration in m6A Regulators in EMs

The m6A regulators investigated in this study were derived from previous findings [34,35]. The landscape of the expression and correlation of 23 m6A regulators, including 8 writers, 13 readers, and 2 erasers, was assessed in EM lesions, the in situ endometrium, and the normal endometrium, while the expressed differences in these regulators were compared comprehensively. The protein–protein interaction (PPI) network of m6A regulators was obtained from the STRING database (https://string-db.org/, accessed on 11 November 2022).

### 4.3. Alteration of Differentially Expressed mRNAs Associated with m6A Regulators

Biological databases with a wide variety of human protein interaction networks are emerging as a result of ongoing research on the roles of human proteins. Here, five databases that provide a more comprehensive view of human protein interactions verified by different essays and experimental methods, namely HPRD (Human Protein Reference Database, http://hprd.org/index_html, accessed on 1 December 2022), BIND (Biomolecular Interaction Network Database, http://bind.ca/, accessed on 1 December 2022), MINT (Molecular INTeraction Database, http://mint.bio.uniroma2.it/mint/, accessed on 1 December 2022), IntAct (IntAct Molecular Interaction Database, http://www.ebi.ac.uk/intact/index.html, accessed on 1 December 2022), and DIP (Database of Interacting Proteins, http://dip.doe-mbi.ucla.edu/, accessed on 1 December 2022), were applied. Based on these background networks, the nearest neighbor networks composed of m6A regulators and mRNAs that were confirmed to interact with m6As (m6A_PPI) were filtered out, from which the differential expression of the key m6A-related mRNAs in EMs was extracted. Pearson correlation analysis was performed on these key mRNAs and m6As, and each mRNA–m6A pair with an absolute correlated coefficient value >0.3 and *p* < 0.05 was selected to construct the mRNA and m6A co-expression network (MACN), while the m6A nodes overlapping with m6A_PPI were labeled as mRNA-related m6As. Moreover, the functional roles of the related mRNAs were integrated as enriched terms or pathways.

### 4.4. Consensus Clustering Analysis and Immune Microenvironment Characteristics in EMs

Firstly, based on the expression of the screened m6A_PPI, an unsupervised clustering method was performed to identify heterogeneous patterns of mRNA modifications, conducted using the ConsensusClusterPlus package on all EM samples, while the number of clusters was evaluated through iterations to ensure a robust classification. The determined optimal number of clusters was based on a cumulative distribution function, and variations in clinical characteristics across subtypes were assessed further. Then, the relationship between the unsupervised classification and infiltrated immune cells was explored using samples with an empirical CIBERSORT *p*-value < 0.05. Additionally, the scores of different subtypes were assessed using an R package estimate to evaluate the abundance of immune or stromal infiltration in tissues, as well as estimate scores tested by the Wilcoxon rank sum test.

### 4.5. Identification of m6A-Related lncRNAs

Using the reference genome GRCh38, which contains 17,944 lncRNAs, a lncRNA expression matrix for EM was constructed, and all lncRNAs were examined for differential expression among different conditions, with FDR < 0.05 considered as a significant threshold. Next, a co-expression network (lncRNA and m6A co-expression network, LACN) composed of differentially expressed lncRNAs in the previous step was created based on Pearson correlation analysis, setting the criteria to an absolute correlated coefficient value of greater than 0.3 and *p* < 0.05. An edge can be joined to a significant lncRNA–m6A pair; thus, the network can be cross-linked. The key lncRNA-associated m6As (labeled lncRNA-related m6As) that regulate EM with a more remarkable score than the mean of all node scores were chosen using the PageRank method, based on the calculation of the igraph package.

### 4.6. Establishment and Validation of m6A Diagnostic Model for EMs

For the lncRNA-related m6As and mRNA-related m6As explored above, their common parts were extracted as training parameters. The least absolute shrinkage and selection operator (LASSO) regression-based method was used with 10-fold cross-validation for continuous shrinkage among the input variables, and the selected features were considered as diagnostic indicators for patients with EM using the glmnet package, which were further confirmed to have the most significant impact on the EM prediction by the receiver operating characteristic (ROC) curves. A stepwise method with a “both” mode was then performed and verified by testing multicollinearity for a more compact model. The rms package was used to perform and validate the model, and the effect of these key m6As was visualized using a nomogram plot. Then, the diagnostic model was validated using independent GSE86534 and GSE105764 cohorts.

### 4.7. Clinical Sample Collection

Patients with ovarian EM who underwent surgery at the Second Affiliated Hospital of Harbin Medical University from July 2021 to July 2022 were enrolled, all classified as stage III–IV according to the revised American Fertility Society (AFS-r) classification. The eutopic endometrium tissues (EU) and ovarian endometriosis tissues (EC) were collected in a total of 12 cases, of which the paired specimens were exactly matched. Moreover, 12 cases of normal control endometrium tissues (NM) diagnosed as cervical lesions were collected. Specimens in all 3 groups were detected to be in the proliferative phase in the menstrual cycle by postoperative pathology, excluding hormone treatment for nearly six months. All tissues were verified by two independent experienced histopathologists.

### 4.8. Primary Cell Extraction

The tissue specimens were rinsed with saline and twice with PBS, and then cut into a paste and transferred to a culture dish with collagenase type IV (1 mg/mL). The culture dish was placed in a 37 °C incubator for 2 h with gentle shaking, and then the tissue debris and other cells, such as endometrial epithelial cells, were removed with a 40 mm sieve. The filter was placed in a centrifuge for 10 min at 1000 r/min, and the supernatant was aspirated to obtain cell precipitates. The cells were resuspended by adding complete DMEM/F12 culture medium to the centrifuge tube and transferred to culture flasks.

### 4.9. Reverse Transcription and RT-qPCR

Total RNA from ovarian EM of patients and endometrial stromal cells was extracted with TRIzol reagent (Ambion, USA) and converted to complementary DNA using a PrimeScript™ RT reagent Kit with gDNA Eraser (Takara, Japan). RT-qPCR was performed with TB Green^®^ Premix Ex Taq™ (Takara, Japan). The settings were as follows: 40 cycles of 15 min at 37 °C, 5 s at 60 °C, and 30 s at 72 °C. All relative mRNA expression levels were analyzed using the 2^−ΔΔCt^ method.

The primers used are listed in Appendix A.

### 4.10. m6A Quantity Assay

The relative levels of m6A were measured using an m6A RNA Methylation Quantification Kit (Colorimetric) (Epigentek, USA). RNA was extracted using the TRIzol method, as mentioned before, and added to a 96-well plate according to the manufacturer’s instructions. Following the instructions, RNAs were well-bonded to the strips at 37 °C for 90 min with binding solution. After adding capture and detect solution, the m6A levels were read at a wavelength of 450 nm. The data were calculated using relative quantification with three repeat wells obtained from each reaction.

### 4.11. Western Blotting

The harvested cells were washed with cold PBS and then lysed with RIPA buffer containing added PMSF on ice for 30 min. The lysate was centrifuged at 12,000 rpm for 10 min at 4 °C, and the supernatant was collected. The total protein concentration was determined using a BCA protein assay kit (meilunbio). A moderate amount of protein (20 μg) of each sample was separated by SDS-PAGE and transferred to PVDF membranes. After being closed with fast closure solution for half an hour, membranes were incubated with anti-METTL3 (1:2000, Abcam), YTHDF2 (1:5000, Proteintech), FTO (1:1000, Proteintech) and GAPDH (1:5000, Proteintech) at 4 °C overnight. The membranes were then washed 3 times with TBST and incubated with horseradish peroxidase (HRP)-labeled goat anti-rabbit secondary antibody (1:5000, Bioss) for 1 h at room temperature. The results of the strips were observed using the Enhanced Chemiluminescence Detection Kit (Meilunbio, Dalian, China).

### 4.12. Statistical Analysis

Comparisons were analyzed using the Wilcoxon rank sum test for two groups or the Kruskal–Wallis test for more than two groups. The false discovery rate (FDR) correction method was applied to the *p*-values in the differential expression analysis. Here, log2FC > 0 was considered an upregulated gene, as described previously, while log2FC < 0 was considered a downregulated gene [36,37]. All experiments were repeated three times or more, and all statistical and visualization work was carried out using GraphPad Prism 9.0 and SPSS software. The results are presented as the mean ± standard deviation. A comparison of all experimental results between two groups was performed using Student’s t-test, or one-way ANOVA among three or more groups. The difference was considered significant at *p* < 0.05 (ns, *p* ≥ 0.05; *, *p* < 0.05; **, *p* < 0.01; ***, *p* < 0.001; ****, *p* < 0.0001).

## 5. Conclusions

In conclusion, with the investigation of the m6A modification patterns in endometriosis, we explored the mRNAs and lncRNAs related to m6A regulators using the GEO database. Two molecular subtypes were identified with different infiltration levels of immune microenvironment cells, which related to the clinical features. We constructed a diagnostic m6A signature of endometriosis and found that METTL3 and YTHDF2 might be the key m6A targets of EM through experiments. Still, only a limited mechanism of METTL3–m6A–YTHDF2 in endometriosis was studied in this paper, and thus subsequent experiments are needed to verify our research results. Our analysis results might be relatively specific to our study conditions; however, they generate new views for the diagnosis and treatment of EM in the future.

## Figures and Tables

**Figure 1 ijms-24-01665-f001:**
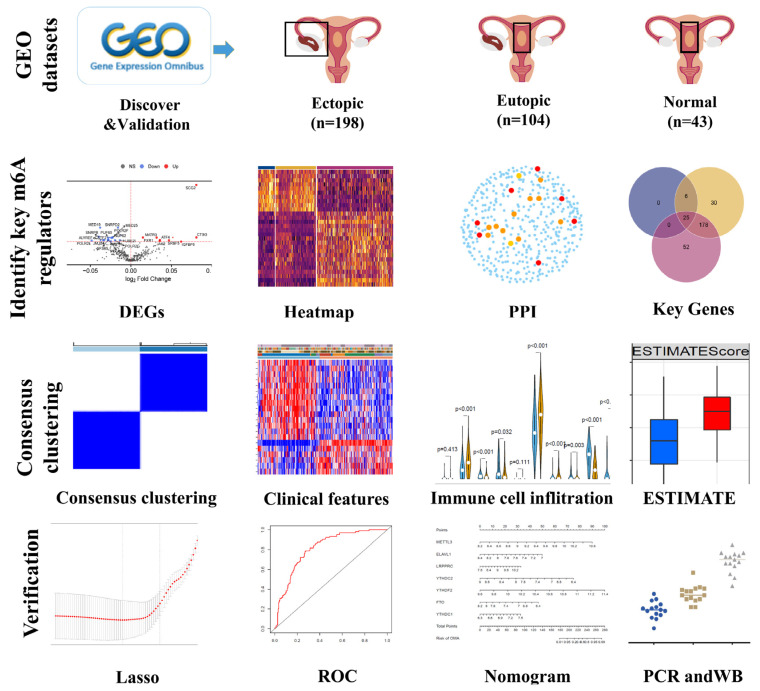
The entire workflow used in this study.

**Figure 2 ijms-24-01665-f002:**
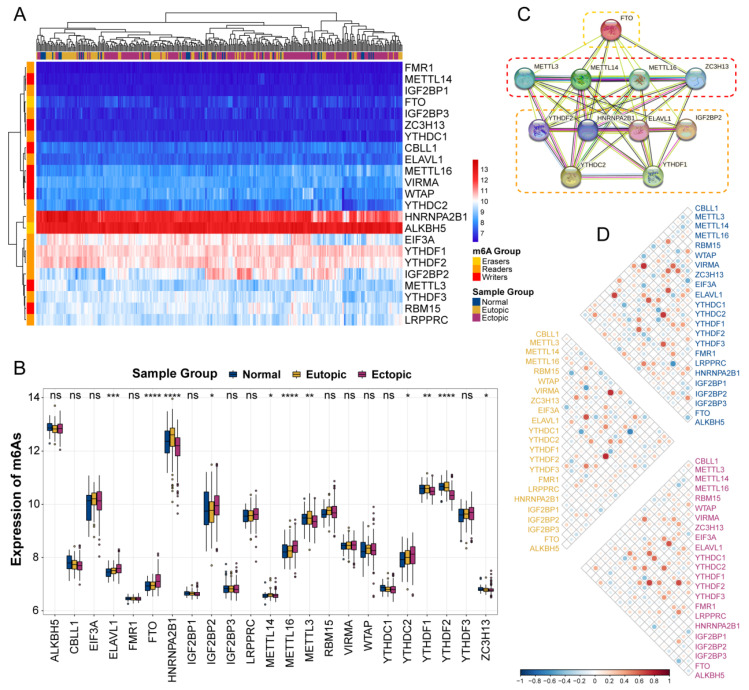
The transcriptome expression status of all m6A regulators between normal, eutopic, and ectopic samples by heatmap (**A**) and boxplot (**B**). (**C**) The protein–protein interactions among differential expressed m6A regulators. Three frames from top to bottom represent erasers, writers, and readers, respectively. (**D**) The correlation analysis of m6A regulators in normal (top), eutopic (median), and ectopic (bottom) groups. ns, *p* ≥ 0.05; *, *p* < 0.05; **, *p* < 0.01; ***, *p* < 0.001, ****, *p* < 0.0001.

**Figure 3 ijms-24-01665-f003:**
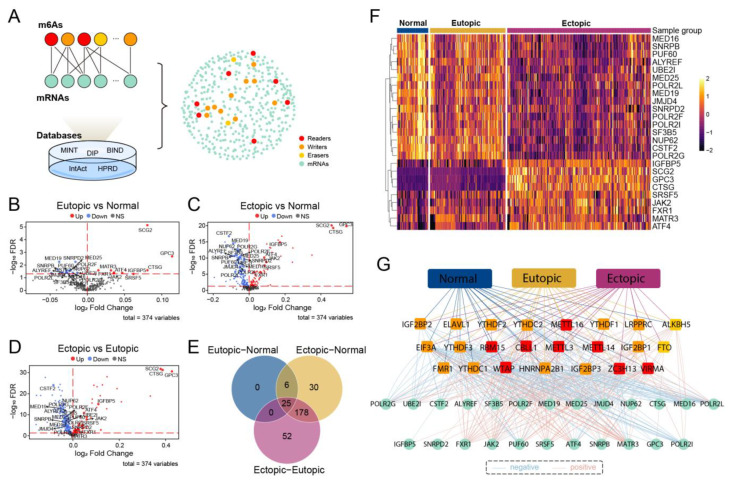
Analysis of m6A regulators-related mRNAs. (**A**) A comprehensive network was extracted from m6A regulators and their nearest neighbor mRNAs in five experimentally validated databases. (**B**–**D**) The volcano plots showed significant variations in these related mRNAs in eutopic vs. normal, ectopic vs. normal, and ectopic vs. eutopic groups. The upregulated genes marked in red, the downregulated in blue and gray indicated not differential expressed. The overlapping mRNAs among 3 differential results were exhibited in the Venn plot (**E**), and their expression in the heatmap (**F**). (**G**) A robust MACN consisted of m6A regulators with interacting mRNAs in normal, eutopic and ectopic groups. The blue lines represented negative correlations, and red ones pointed to the positive.

**Figure 4 ijms-24-01665-f004:**
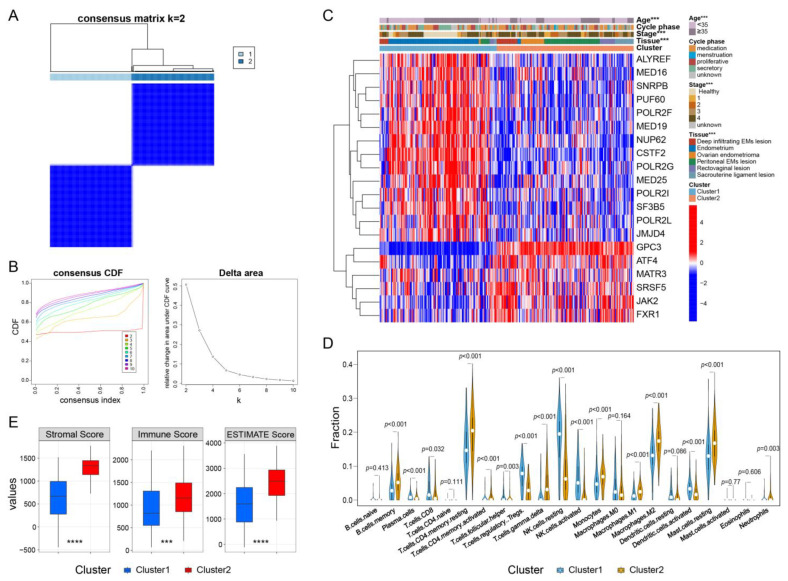
Heterogeneity patterns and immune characteristics of m6A-related mRNAs in EMs. (**A**,**B**) According to the clustergram and delta area plot, the best clustering number was set up to 2. (**C**) Differences in clinicopathologic features and the expression levels of m6A-related mRNAs between the two distinct clusters. (**D**) The infiltration levels of 22 immune cells in the two clusters. (**E**) Different scores (stromal score, immune score, and estimate score) distinguished the configuration and function evidently from clusters. ***, *p* < 0.001, ****, *p* < 0.0001.

**Figure 5 ijms-24-01665-f005:**
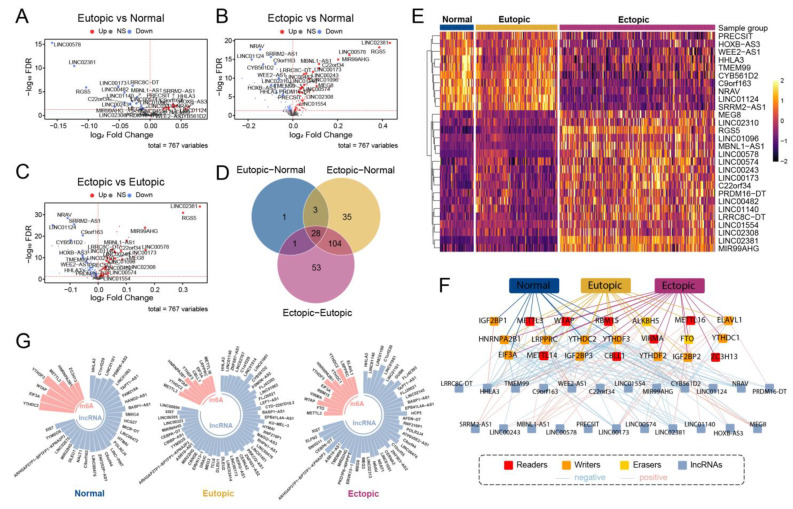
Analysis of m6A regulators-related lncRNAs. (**A**–**C**) The volcano plots showed significant variations in these related lncRNAs in eutopic vs. normal, ectopic vs. normal, and ectopic vs. eutopic groups. The upregulated genes marked in red, the downregulated in blue and gray indicated not differential expressed. The overlapping lncRNAs among 3 differential results were exhibited in the Venn plot (**D**), and their expression in the heatmap (**E**). (**F**) A robust LACN consisted of m6A regulators with interacting lncRNAs in normal, eutopic and ectopic groups. (**G**) Random walk algorithm determined key nodes in normal, eutopic and ectopic groups, respectively, using Rose Charts.

**Figure 6 ijms-24-01665-f006:**
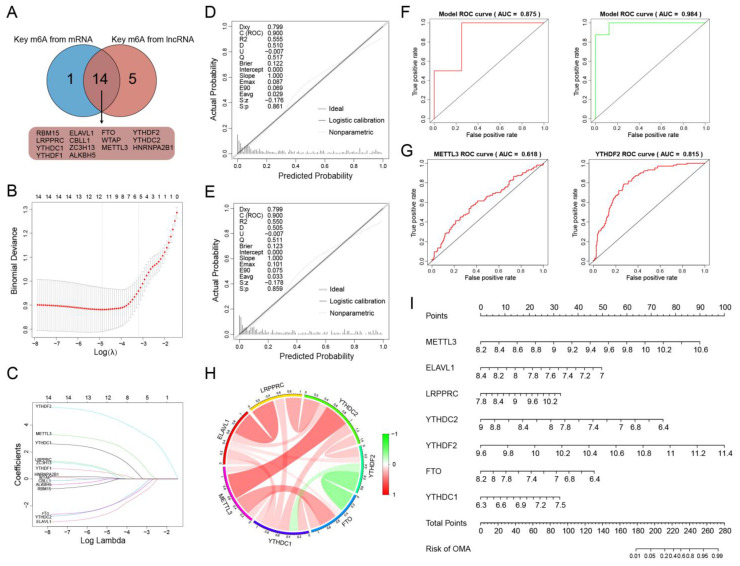
Establishment and validation of m6A diagnostic model. (**A**) The key m6A regulators overlapping in two networks were extracted for model training. 10-fold cross-validation for shrinking parameters (**B**) and extracting non-zero coefficients (**C**). (**D**) A diagnostic model was constructed by 14 m6As for EMs. (**E**) A concise model was further constructed by 7 m6As with AUC remaining high. (**F**) The roc plots for validation of the diagnostic model using GSE86534 and GSE105764. (**G**) The roc plots for METTL3 and YTHDF2, with the most prognostic potentialities among writers and readers, respectively. (**H**) The mutual relationship among the composition of the diagnostic model in circos plot in ectopic samples. (**I**) The nomogram for risk assessment in EMs. ROC, receiver operating characteristic; AUC, area under curve.

**Figure 7 ijms-24-01665-f007:**
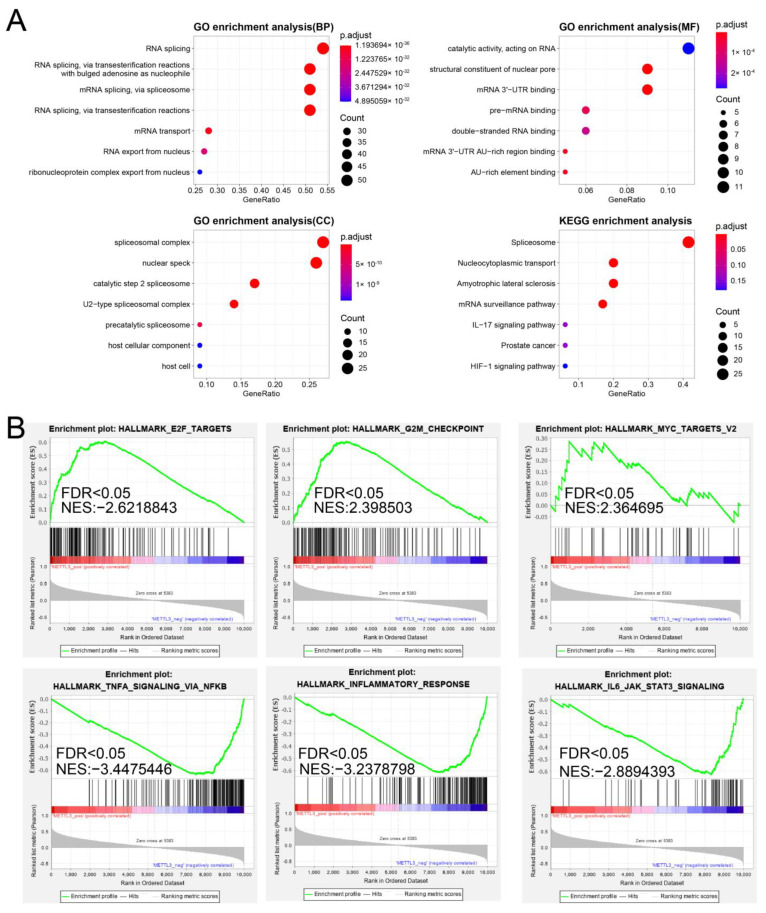
The biological function of the modification patterns of METTL3. (**A**) Top results of enrichment analysis. The gene ontology (GO) terms were under three categories, named biological processes (BP), cell components (CC), molecular functions (MF). KEGG, Kyoto Encyclopedia of Genes and Genomes KEGG pathways. (**B**) The top 3 of significantly positive and negative HALLMARK terms and normalized enrichment scores (NES) in the METTL3-m6A modification pattern in GSEA. FDR, false discovery rate., GSEA, gene set enrichment analysis.

**Figure 8 ijms-24-01665-f008:**
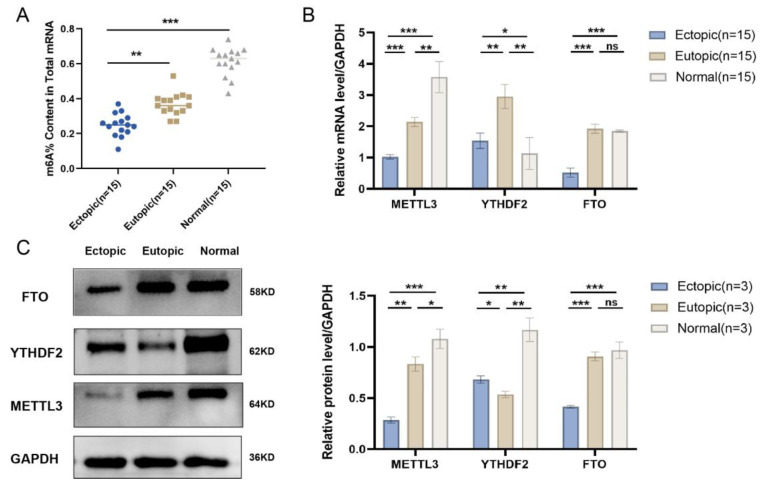
Experimental validation of key m6A regulators in tissues. (**A**) Each m6A level detected by m6A quantity assay in Ectopic, Eutopic and Normal groups. (**B**,**C**) The mRNA expression levels by RT-qPCR (**B**) and protein expression levels by WB (**C**) of METTL3, YTHDF2 and FTO in Ectopic, Eutopic and Normal groups. These results were presented as the mean ± SDs. ns, *p ≥* 0.05; *, *p* < 0.05; **, *p* < 0.01; ***, *p* < 0.001.

**Table 1 ijms-24-01665-t001:** Clinical characteristics in GSE141549 (n = 345).

	Ectopic N = 198	Eutopic N = 104	Normal N = 43
**Cycle phase** **:**			
medication	96 (48.5%)	43 (41.3%)	10 (23.3%)
menstruation	10 (5.05%)	7 (6.73%)	0 (0.00%)
proliferative	29 (14.6%)	17 (16.3%)	7 (16.3%)
secretory	41 (20.7%)	27 (26.0%)	17 (39.5%)
unknown	22 (11.1%)	10 (9.62%)	9 (20.9%)
**Tissue:**			
Deep infiltrating endometriosis lesion	42 (21.2%)	0 (0.00%)	0 (0.00%)
Endometrium	0 (0.00%)	104(100%)	43 (100%)
Ovarian endometrioma	28 (14.1%)	0 (0.00%)	0 (0.00%)
Peritoneal endometriosis lesion	79 (39.9%)	0 (0.00%)	0 (0.00%)
Rectovaginal lesion	22 (11.1%)	0 (0.00%)	0 (0.00%)
Sacrouterine ligament lesion	27 (13.6%)	0 (0.00%)	0 (0.00%)
**Stage:**			
1	20 (10.1%)	15 (14.4%)	0 (0.00%)
2	26 (13.1%)	14 (13.5%)	0 (0.00%)
3	47 (23.7%)	22 (21.2%)	1 (2.33%)
4	101 (51.0%)	52 (50.0%)	0 (0.00%)
Healthy	0 (0.00%)	0 (0.00%)	42 (97.7%)
unknown	4 (2.02%)	1 (0.96%)	0 (0.00%)
**Age:**			
<35	135 (68.2%)	68 (65.4%)	5 (11.6%)
≥35	63 (31.8%)	36 (34.6%)	38 (88.4%)

## Data Availability

We are ready to provide details regarding this if published.

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
