# Peer review of "Cross-Talk between N6-Methyladenosine and Their Related RNAs Defined a Signature and Confirmed m6A Regulators for Diagnosis of Endometriosis"

_ijms, 2023, doi:10.3390/ijms24021665_

Round 1

Reviewer 1 Report

The manuscript is complete and well presented. The findings are significant for the diagnosis of endometriosis. Before publication, here are a few minor suggestions:

1. For Figure 3G and Figure 5F, it is not clear how the coefficiency represented by the darkness of the edge.

2. There is a discrepancy in the description of Figure 8B on line 389 in the context and the figure.

3. In Figure 8A, the authors demonstrated a decreased level of m6A in ectopic and eutopic tissues compared to normal tissue. Additionally, the authors showed a decreased m6A writer METTL3 level in ectopic and eutopic tissues. It would be interesting to determine if the increase of the eraser FTO can be validated.

4. It would be interesting to validate the expression changes in RNA/protein levels of some of the m6A interacting genes found in this paper, such as SCG2 and GPC3, by knocking out METTL3, YTHDF2, and FTO. This would provide strong evidence to demonstrate that m6A regulators may affect the development of endometriosis by regulating the expression of key genes through the regulation of m6A levels.

Author Response

Dear editors and reviewers,

I’m writing to thank you very much for returning to us the reviews of our manuscript entitled “Cross-Talk Between n6-Methyladenosine and Their Related RNAs Defined a Signature and Confirmed m6A Regulators for Diagnosis of Endometriosis” (ID: ijms-2134749). This paper was submitted to your journal on the 16 December 2022. We very much appreciate all of the comments received in review.

We have studied the comments carefully and have made revision marked in red accordingly. The main corrections in the paper and the responds to the reviewer’s comments are as follow profile.

Once again, thank you very much for your serious review and constructive comments which would help us both in English and in depth to improve the quality of the paper.

Kind regards,

Xiaotong Wang, Guangmei Zhang

Reviewer 2 Report

1.The abstract of the article is not clearly described in many places and only states the methods used and the biological implications they represent are lacking, For example,

a. In page1 line 13, remarkable chosen by PageRank method”. The PageRank algorithm is a network-based method for ranking the importance of nodes, and in this study the algorithm is applied to what network. The current statement is confusing,

b.Which features to perform k-mean clustering analysis based on need to be clearly stated in the abstract, is it lncRNA, mRNA or m6A?

c. What kind of results did Cibersort get.

d.What features were used for the construction of the m6A correlation risk model

The abstract is not sufficient to reflect the methodology of the study and the results obtained and could be recondensed.

2. In Figure1, step2 Indentify key m6A regulation. PageRank is before PPI, is there a process order error here? PageRank does not appear in the results, so what results are obtained by remarkable chosen based on PageRank.

3. What was the basis for the choice of age in Table1, using 35 as the cut-off point?

4. Page 5 The full phrase “MACN” should mention at the first occurrence.

5. In 2.3 “410 mRNAs interacting with 19 m6As were discovered as one-step neighbors in 5 public databases”, 5 public database constitutes a protein interaction network, one-step neighbors are screened based on the protein interaction network, what methods and software are used here.

6. In manuscript, “410 mRNAs interacting with 19 m6As were discovered as one-step neighbors in 5 public databases(Figure 3A), with 374 mRNAs included here for analyzing their expressed alteration separately. Three gene groups were obtained, that is, 31 mRNAs in the eutopic vs normal group(Figure 3B), 239 mRNAs in the ectopic vs normal group(Figure 3C), and 255 mRNAs in the ectopic vs eutopic group(Figure 3D), with each FDR<0.05.” These statement is confusing.

7. In 2.4, “The value of k = 2 was assessed as the most appropriate number of clusters for further analysis according to the delta area plot and matrix heatmap and then referred them as cluster1 and cluster2 respectively(Figures 4A-B, Supplementary Figure 1, Supplementary Figure 2).“ What is the reason for choosing k=2?

8. In 2.6. ”767 lncRNAs were screened from the reference genome for differentially expressed analysis“. Which version of the reference genome is?

9. The full phrase “LACN” should mention at the first occurrence.

10. In page 7 line 184, “Furthermore, the random walk algorithm was applied to determine the key m6As”. PageRank is one of the random walk algorithm, In this part of the study is PageRank used or not. This result is not consistent with the flow of Figure1

11. In 2.8, only METTL3 is discussed in 2.8, but YTHDF2 possessed the highest risk weight, YTHDF2 related modification patterns should be researched as well.

12. 6. Patents“ is not necessary

There are no figure legends for Figure S1-S3.

Author Response

Dear editors and reviewers,

I’m writing to thank you very much for returning to us the reviews of our manuscript entitled “Cross-Talk Between n6-Methyladenosine and Their Related RNAs Defined a Signature and Confirmed m6A Regulators for Diagnosis of Endometriosis” (ID: ijms-2134749). This paper was submitted to your journal on the 16 December 2022. We very much appreciate all of the comments received in review.

We have studied the comments carefully and have made revision marked in red accordingly. The main corrections in the paper and the responds to the reviewer’s comments are as follows profile.

Once again, thank you very much for your serious review and constructive comments which would help us both in English and in depth to improve the quality of the paper.

Kind regards,

Xiaotong Wang, Guangmei Zhang

Reviewer 3 Report

With the datasets from GEO database, the authors performed a comprehensive assessment using various bioinformatic tools to investigate the correlation between m6A regulators and mRNAs as well as lncRNAs. Based on the mRNAs and m6A co-expression network (MACN) and the lncRNAs and m6A co-expression network (LACN), the authors constructed a m6A-related diagnostic signature and identified two most relevant m6A regulators METTL3 and YTHDF2, which could be diagnostic targets of endometriosis. However, there are some issues/concerns with the manuscript.

Issues:

Page 6, bottom: “in the ectopic group, erasers had higher expression while writers had significantly downregulated expression compared to the normal group” According to Figure 2B, only the eraser FTO was significantly upregulated and the methyltransferase complex METTL3 and METTL14 were significantly downregulated. The changes of eraser ALKBH5 and writer METTL16 were not consistent with the statement here.

Page 7, bottom: “410 mRNAs interacting with 19 m6As…” Does “m6As” in the context refer to m6A regulator? Authors should clearly distinguish m6A, m6A regulators, mRNA, m6A-associated mRNA in the manuscript to avoid confusion for readers.

Page 8, bottom: According to the CDF curve and delta area plot in Figure 4B, k = 7 should be the most appropriate number of clusters for further analysis. When k equals to 7, the clustering results are relatively stable and the area under the CDF curve starts to remain stable. Authors should explain why they consider k=2 as the most appropriate number of clusters.

Page 10, top: Authors used random walk algorithm to find the key m6A regulators among lncRNA-m6A regulator pairs. Why didn’t authors use random walk algorithm to determine the key m6A regulators of the m6A regulators and mRNA pairs in NM, EU, EC groups, which could be a supplementary support of their findings?

Page 11 top: According to the predicted accuracies of seven m6A regulators, there should be three most relevant regulators: the only methyltransferase METTL3 (AUC = 0.618), the only demethylase FTO (AUC = 0.659), and the reader YTHDF2 (AUC = 0.815) with highest AUC values. However, the authors stated that “…METTL3 had the highest AUC value among all writers and YTHDF2 had…” and only considered METTL3 and YTHDF2 as the most relevant regulators. More reasonable explanations should be given. Or, the hypothesis that the METTL3-m6A-mRNA/lncRNA-YTHDF2 axis plays a vital role in the progression of Ems is not convincing.

Page 12: Enrichment analysis did not provide a lot of useful information and is not high correlated with the objective of the manuscript, especially for the GO analysis.

Page 13, top: The results of RT-qPCR (Figure 8B) showed that the level of YTHDF2 mRNA in normal group was lower than that in ectopic group, while the authors stated that “The qRT-PCR experiment revealed that Ectopic and Eutopic group had considerably lower METTL3 and YTHDF2 mRNA levels than Normal”, which is confusing.

Figure 2C: The correlation score was not only represented in color but also in size. Authors should describe this in detail in the legend.

Figure 4C: Why were these 20 mRNAs chosen for cluster and expression analysis? At the bottom of Page 7, the authors indicated that “…a co-expressed network in the NM group was comprised 256 of 20 m6As and 21 mRNAs, of which 19 m6As and 24 mRNAs, 19 m6As and 22 mRNAs 257 in the EU and the EC group respectively.” Are these 20 mRNAs an overlap of the three groups? Why didn’t the authors use the 25-overlapping mRNA shown in Figure 3E?

Figure 8C: Replicate numbers of western blot should be displayed.

Mino issues:

The title: I would recommend changing Cross-Talk Between n6-Methyladenosine and Their Related RNAs Defined a Signature and Confirmed m6A Regulators for Diagnosis of Endometriosis to Cross-Talk Between N6-Methyladenosine and Their Related RNAs Defined a Diagnostic Signature for Endometriosis with m6A Regulators as Diagnostic Targets

Page 2, top: “cytosine hydroxylation” should match with “hm5C”.

Page 10, top: Change “….scored more remarkable in all three groups, hence might be a…” to “….scored more remarkably in all three groups and thus might be a…”

Page 13, top: Please change “qRT-PCR” to “RT-qPCR”.

Page 15, conclusion part: I would recommend removing “…such as Me-rip sequencing to yield the most quantification accuracy and insightful mechanisms from plural perspectives, which is undergoing tests based on these key m6A regulators obtained here.”, which is redundant and unnecessary.

Figure 3B, 3C, 3D, 5A, 5B, 5C:  Annotations overlap with each other in the graphs. I recommend removing some unnecessary annotations and keeping significant ones for readers to get the key information.

Author Response

(The authors gave the same response as above.)
